# DYRK1A signalling synchronizes the mitochondrial import pathways for metabolic rewiring

Adinarayana Marada[1,10], Corvin Walter[1,2,10], Tamara Suhm[1], Sahana Shankar[1], Arpita Nandy[1,2,3], Tilman Brummer [4,5,6], Ines Dhaouadi[1], F.-Nora Vögtle [7,8,9] ✉ & Chris Meisinger [1,5,9] ✉

Mitochondria require an extensive proteome to maintain a variety of metabolic reactions, and changes in cellular demand depend on rapid adaptation of the mitochondrial protein composition. The TOM complex, the organellar entry gate for mitochondrial precursors in the outer membrane, is a target for cytosolic kinases to modulate protein influx. DYRK1A phosphorylation of the carrier import receptor TOM70 at Ser91 enables its efficient docking and thus transfer of precursor proteins to the TOM complex. Here, we probe TOM70 phosphorylation in molecular detail and find that TOM70 is not a CK2 target nor import receptor for MIC19 as previously suggested. Instead, we identify TOM20 as a MIC19 import receptor and show off-target inhibition of the DYRK1A-TOM70 axis with the clinically used CK2 inhibitor CX4945 which activates TOM20-dependent import pathways. Taken together, modulation of DYRK1A signalling adapts the central mitochondrial protein entry gate via synchronization of TOM70- and TOM20-dependent import pathways for metabolic rewiring. Thus, DYRK1A emerges as a cytosolic surveillance kinase to regulate and fine-tune mitochondrial protein biogenesis.

Mitochondria are central hubs of homeostasis and metabolism in eukaryotic cells. More than 1000 (yeast) and 1500 (human) different proteins carry out the plethora of metabolic reactions located in these organelles[1–7] and appr. 99% of these mitochondrial proteins have to be imported as precursor proteins from the cytosol. The translocase of the outer membrane (TOM complex) serves as central entry gate for virtually all precursor proteins. A stable TOM core complex consisting of the translocation pore TOM40, the central receptor TOM22 and three small TOM subunits dynamically interacts with two peripheral

import receptors, TOM20 and TOM70[8–10]. TOM20 preferentially binds precursor proteins with N-terminal presequences that are further transferred to the TIM23 translocase in the inner membrane. TOM70 is the major import receptor for precursors of the metabolite carrier family that carry internal targeting information and are handed over to the TIM22 translocase after their passage across the outer membrane and intermembrane space. In addition to these two major import pathways, several other sorting machineries have been identified, e.g. the SAM complex for β-barrel assembly in the outer membrane or the

[1]Institute of Biochemistry and Molecular Biology, ZBMZ, Faculty of Medicine, University of Freiburg, 79104 Freiburg, Germany. [2]Faculty of Biology, University of Freiburg, 79104 Freiburg, Germany. [3]Spemann Graduate School of Biology and Medicine, University of Freiburg, 79104 Freiburg, Germany. [4]Institute of Molecular Medicine, ZBMZ, Faculty of Medicine, University of Freiburg, 79104 Freiburg, Germany. [5]BIOSS Centre for Biological Signalling Studies, University of Freiburg, 79104 Freiburg, Germany. [6]German Cancer Consortium DKTK Partner Site Freiburg, German Cancer Research Center (DKFZ), Heidelberg, Germany. [7]Center for Molecular Biology of Heidelberg University (ZMBH), DKFZ-ZMBH Alliance, 69120 Heidelberg, Germany. [8]Network Aging Research, Heidelberg University, 69120 Heidelberg, Germany. [9]CIBSS - Centre for Integrative Biological Signalling Studies, University of Freiburg, 79104 Freiburg, Germany. [10]These authors contributed equally: Adinarayana Marada, Corvin Walter ✉ e-mail: n.voegtle@zmbh.uni-heidelberg.de; chris.meisinger@biochemie.uni-freiburg.de

MIA machinery which traps proteins with characteristic twin $CX_9C$ motifs by oxidative protein folding in the intermembrane space (IMS)[7,11–17].

The mitochondrial import machineries have long been regarded as static and constitutively active entities, and it was largely unknown how the organellar proteome can be dynamically shaped upon changes in metabolic demand. Only recently, the TOM complex has been discovered as a target of cytosolic protein kinases, allowing the regulation of protein influx upon metabolic rewiring or during different cell cycle phases[18–21]. While most of the signaling cascades and their target sites at the TOM complex have been identified in the yeast model system, knowledge about similar regulation in mammalian mitochondria remained scarce.

In a global and unbiased profiling of the human kinome, we identified DYRK1A as the first kinase of the mammalian mitochondrial import machinery. DYRK1A is a member of the dual specificity tyrosine-regulated kinase family and phosphorylates the carrier import receptor TOM70 at position Ser91 in human and Ser94 in mouse mitochondria[22]. DYRK1A-dependent phosphorylation stimulates the molecular interaction of TOM70 with the TOM core complex. This allows efficient transfer of TOM70-bound carrier precursors to the import pore. The resulting increased import rate of metabolite carrier proteins ultimately stimulates respiratory activity[22].

The activation of the carrier import pathway via the DYRK1A-TOM70 signaling axis was challenged by a recent publication, which proposes that Ser94, the homologous site of murine TOM70 is phosphorylated by Casein Kinase 2 (CK2) and that this phosphorylation plays an inhibitory role in the import of the IMS protein MIC19[23]. MIC19 is a central component of the mitochondrial contact site and cristae organizing system (MICOS), a modular protein machinery that is mainly located at the cristae junctions in the inner membrane and important for shaping of the cristae structure and thus respiratory activity. MIC19, that harbors the typical twin $CX_9C$ motif of MIA substrates, plays an important role in the biogenesis of MICOS and coordinates multiple interactions of MICOS with other protein complexes[24–30]. Latorre-Muro et al. implicate TOM70 as a central regulatory hub to control mitochondrial cristae formation by modulating MIC19 import and thus MICOS structure and function[23]: cold stress or β-adrenergic stimulation induces the PERK-OGT signaling cascade and glycosylation of TOM70[Ser94]. This leads to increased import of MIC19, which promotes cristae formation and respiration. The activating PERK-OGT signaling cascade is attenuated by CK2-mediated phosphorylation of TOM70[Ser94], which inhibits MIC19 import. Thus, regulation of MIC19 import via a CK2-TOM70 axis is proposed to link mitochondrial cristae biogenesis and respiration to cold stress and β-adrenergic responses[23,31]. How the inhibitory role of the TOM70[Ser94] phosphorylation site is exerted at the molecular level and why this would only affect the IMS targeted MIC19 precursor, which does not belong to the metabolite carrier family of TOM70-dependent precursor proteins, remained open.

Here, we set out to probe both TOM70 phosphorylation models in molecular detail and to investigate the potential of DYRK1A and CK2 to function as signaling hubs to adapt mitochondrial protein biogenesis pathways and to link protein import to metabolic rewiring. We find that TOM70 phosphorylation is not affected in differentiated brown adipose tissue (BAT) cells upon β-adrenergic stimulation in vivo, and that also MIC19 import is not altered under these conditions in organello. Utilizing several independent experimental approaches and model systems, we show that CK2 is unable to phosphorylate TOM70 and that MIC19 does not require TOM70 as import receptor. Instead, we identify TOM20 as import receptor for MIC19. Intriguingly, we discovered a cross signaling mechanism that provides an explanation for the main contradictory findings that built the base for the two opposing TOM70 phosphorylation models: The CK2 inhibitor CX4945 used by Latorre-Muro et al. displays severe off-target effects and strongly inhibits DYRK1A kinase. Thus, findings that were attributed to impaired CK2 signaling were in effect caused by compromised DYRK1A activity. Here, we show that impaired DYRK1A signaling in turn triggers a protective transcriptional response that leads to profound remodeling of the TOM complex, including increased protein levels of TOM20. As MIC19 import depends on the TOM20 receptor, CK2 inhibition by CX4945 results in increased MIC19 biogenesis via the DYRK1A-TOM70 signaling axis. Intriguingly, we find that this DYRK1A-dependent remodeling of the import machinery stimulates IMS import of further MIA substrates and also activates the classical presequence import pathway.

Taken together, our study uncovers DYRK1A as a regulatory platform that links TOM70- and TOM20-dependent import pathways in mitochondria thus surveilling and fine-tuning mitochondrial protein biogenesis to changes in metabolic demands.

## Results

### TOM70 phosphorylation and MIC19 import are independent of Casein kinase 2

Both proposed models on the regulation of the mammalian TOM complex depend and are built on changes in the Ser91 (human)/Ser 94 (murine) phosphorylation site of TOM70. Therefore, to molecularly probe the two opposing TOM70 phosphorylation models, we first tested whether TOM70 phosphorylation is indeed altered in the used model systems and assessed Tom70[Ser94] levels upon β-adrenergic stimulation of BAT cells in vivo. We generated differentiated mouse BAT cells (Fig. 1a) and incubated them with 1 μM Norepinephrine (NE) for 18 h (refs. 23,32; Methods). We isolated mitochondria from stimulated and mock-treated cells and tested TOM70[Ser94] phosphorylation, which had not been assessed so far[23], by two independent approaches: (i) Phos-tag electrophoresis, which allows the simultaneous detection and quantitative assessment of phosphorylated and non-phosphorylated TOM70, and (ii) immunoblotting using phospho-specific antibodies that recognize phosphorylated TOM70[Ser94] in mouse and TOM70[Ser91] in human samples[22]. We found that TOM70 phosphorylation was almost quantitative and did not change upon NE treatment (Fig. 1b,c). Also endogenous levels of MIC19 and of the MICOS complex (profiled via Blue Native electrophoresis) did not change (Supplementary Fig. 1a, b). To test whether MIC19 import into mitochondria was dependent on NE stimulation, radiolabelled MIC19 precursor protein was generated by in vitro transcription/translation and incubated with isolated mitochondria from NE-treated and mock-treated BAT cells. Import into mitochondria was monitored by Proteinase K protection followed by SDS-PAGE and autoradiography as in ref. 23. We find that import of MIC19 into BAT mitochondria does not depend on stimulation with NE (Fig. 1d, lanes 1–6). We also directly monitored the phosphorylation status of TOM70 by Phos-tag electrophoresis in each import assay and found no changes in TOM70 phosphorylation (Fig. 1d, lanes 7 and 8). In summary, we find that TOM70 is almost quantitatively phosphorylated in mitochondria from BAT cells and that NE stimulation does not alter the level of TOM70 phosphorylation or MIC19 import.

While our analysis of mitochondria from BAT cells did not detect a role for NE in TOM70 phosphorylation and MIC19 import (Fig. 1b–d; Supplementary Fig. 1a, b), we were curious whether MIC19 import could still be modulated by CK2 as proposed by Latorre-Muro et al. [23]. We therefore examined the import of $^{35}$S-labeled MIC19 precursor into isolated mitochondria and found that direct CK2 treatment or in organello inhibition of CK2 by the inhibitor CX4945[23] did not change MIC19 import kinetics (Supplementary Fig. 1c, d). The phosphorylation status of TOM70 was also not affected, which was almost fully phosphorylated under all conditions tested.

Next, we undertook a detailed mechanistic analysis of the molecular role of CK2 in TOM70 phosphorylation. We surveyed the amino acid sequence of the phosphorylation site of human TOM70[Ser91] and

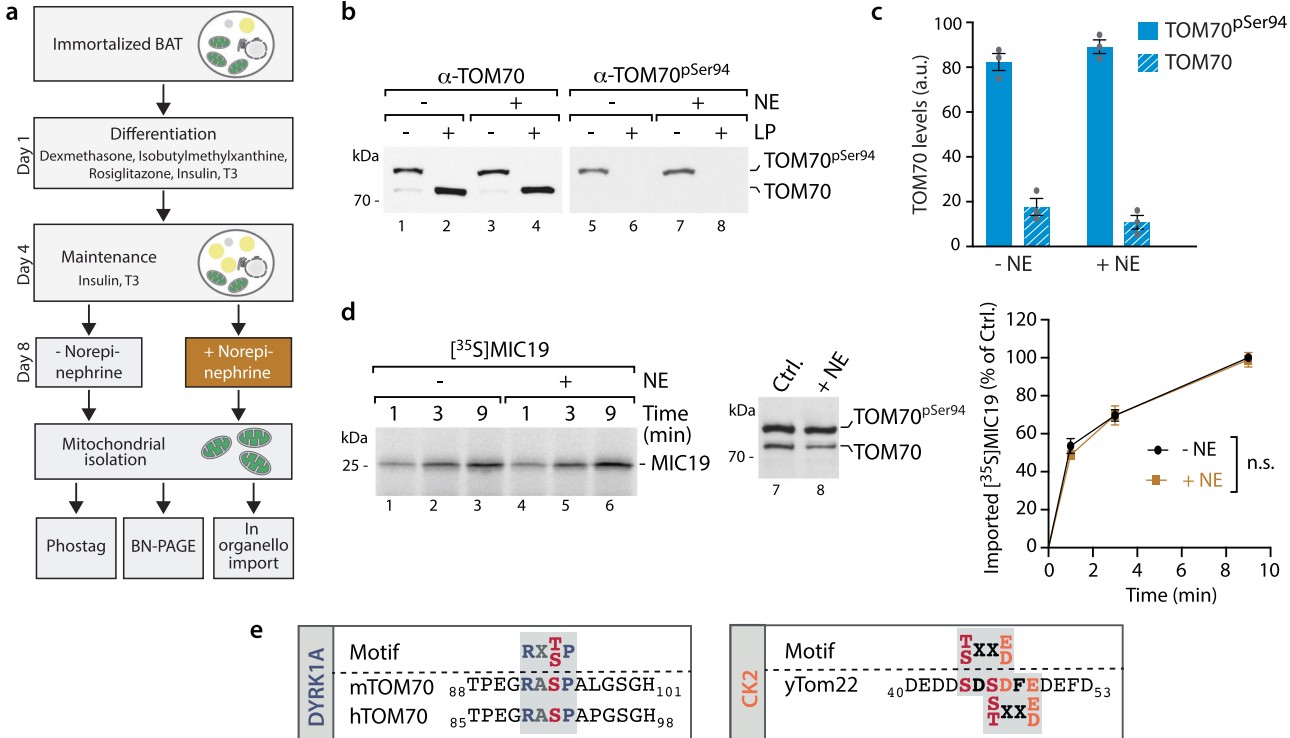

**Fig. 1 | β-adrenergic stimulation of differentiated BAT cells does not change TOM70 phosphorylation and MIC19 biogenesis. a** Schematic representation of the experimental strategy to generate differentiated brown adipocyte tissue (BAT) cells and analyze TOM70 phosphorylation and MIC19 biogenesis. Norepinephrine treatment was for 18 h[23,32]. **b** Mitochondria isolated from differentiated primary immortalized BAT cells incubated in the presence or absence of Norepinephrine (NE) as shown in **a** were analyzed via Phos-tag gels followed by immunodecoration with TOM70 antisera (lanes 1–4) and phospho-specific antibody recognizing TOM70[pSer94] (lanes 5-8). Lambda phosphatase (LP) treatment allows distinction of phosphorylated and non-phosphorylated TOM70. **c** Quantification of phosphorylated and non-phosphorylated TOM70 of samples from **b**, via Phos-tag electrophoresis. Data represent mean ± SEM from three independent experiments. a.u. arbitrary units. **d** Import of [35S]MIC19 precursor protein into mitochondria from **b**. Non-imported precursor was removed by Proteinase K. Samples were analyzed by SDS-PAGE and imported MIC19 was detected by autoradiography. For quantification the control reaction (without NE) at 9 min import time point was set to 100%. Data represent mean ± SEM from three independent experiments. A multiple paired t test with a false discovery rate (FDR) of 1% and a two-stages step-up method of Benjamini, Krieger and Yekutieli was performed to compare between two groups. See the methods section for details on statistical analyses. The TOM70 phosphorylation status was monitored by Phos-tag gels (lanes 7 and 8). n.s. not significant. **e** Profiling of human TOM70[Ser91] and mouse TOM70[Ser94] for kinase motifs reveal consensus sites for DYRK1A. Yeast Tom22 has two consensus sites (Ser44/Ser46) for CK2. Source data are provided as a Source Data file.

mouse TOM70[Ser94]. The consensus motifs recognized and targeted by DYRK1A and CK2 are well characterized[18,33–35]. Inspection of the mammalian TOM70 phosphorylation site reveals a clear consensus motif for DYRK1A[33], but no preference for CK2; see Fig. 1e in which we included the typical CK2 motif found e.g. in yeast Tom22[18].

DYRK1A was identified as TOM70 kinase by a global, unbiased screen of the human kinome (KinaseFinder)[36] and has been validated to phosphorylate human TOM70[Ser91] in vitro, in organello and in vivo[22]. To probe the DYRK1A-TOM70 model, we re-evaluated the screen for the purified TOM70 receptor (cytosolic) domains and now profiled the WT against a TOM70[Ser91Ala] variant[22]. The results clearly show that in effect only kinases of the DYRK family (including CLK and HIPK family members) target TOM70[Ser91], as these kinases were absent in the TOM70[Ser91Ala] variant (Fig. 2a)[22]. Direct comparison of the DYRK family kinases with two active CK2 isoforms (CK2-alpha1 and CK2-alpha2) detected no phosphorylation activity of either CK2 variant, even for the entire WT TOM70 receptor domain (Fig. 2b). Notably, these are high-throughput data testing 245 different human Ser/Thr kinases[22,36]. We therefore sought to validate these results experimentally by applying a direct phosphorylation assay in vitro using the purified cytosolic domains of TOM70[WT] and TOM70[Ser91Ala], and in organello using purified mitochondria. The TOM70 receptor domain was incubated with three different recombinant CK2 variants as well as with DYRK1A and its closely related isoform DYRK1B. Two independent read-outs, Phos-tag electrophoresis and immunoblotting with

phospho-specific antisera against TOM70[pSer91] showed that TOM70[Ser91] can be phosphorylated by DYRK1A and DYRK1B but not by CK2 (Fig. 2c). The CK2 variants tested were active as both phosphorylated the known CK2 target yeast Tom22[cd], which was used as a positive control (Fig. 2c, bottom panel)[18]. In contrast to mammalian TOM70, yeast Tom22 contains the CK2 consensus motif (Fig. 1e).

Next, we analyzed whether CK2 can phosphorylate mitochondrial TOM70: Notably, we tested mitochondria isolated from mouse brain and HEK293T cells, but also from BAT (23; Methods). Phos-tag electrophoresis revealed that TOM70 was almost quantitatively phosphorylated in all samples tested (Fig. 2d, lane 1). Therefore, endogenous phosphorylation had to be removed by lambda phosphatase (LP) prior to kinase assays (Fig. 2d, lane 2)[18,22]. In none of the mitochondria tested CK2 was able to phosphorylate TOM70, whereas DYRK1A/B reproducibly led to a high level of phosphorylation (Fig. 2d, lanes 3–7). Isolated yeast mitochondria were used as positive control to show the activity of CK2 for the known CK2 targets Tom22[Ser44/Ser46] (Fig. 2d, bottom panel)[18,37].

While our results indicate that CK2 cannot target mammalian TOM70, we wanted to test whether changes in TOM70[Ser91] phosphorylation might still impact on MIC19 import. To this end we treated mitochondria with LP and then re-phosphorylated TOM70[Ser91] with DYRK1A (Fig. 2e, lanes 7, 8). However, MIC19 import was not affected (Fig. 2e). In contrast, import of TIM23, a model precursor of the TOM70-dependent carrier import pathway, was stimulated upon

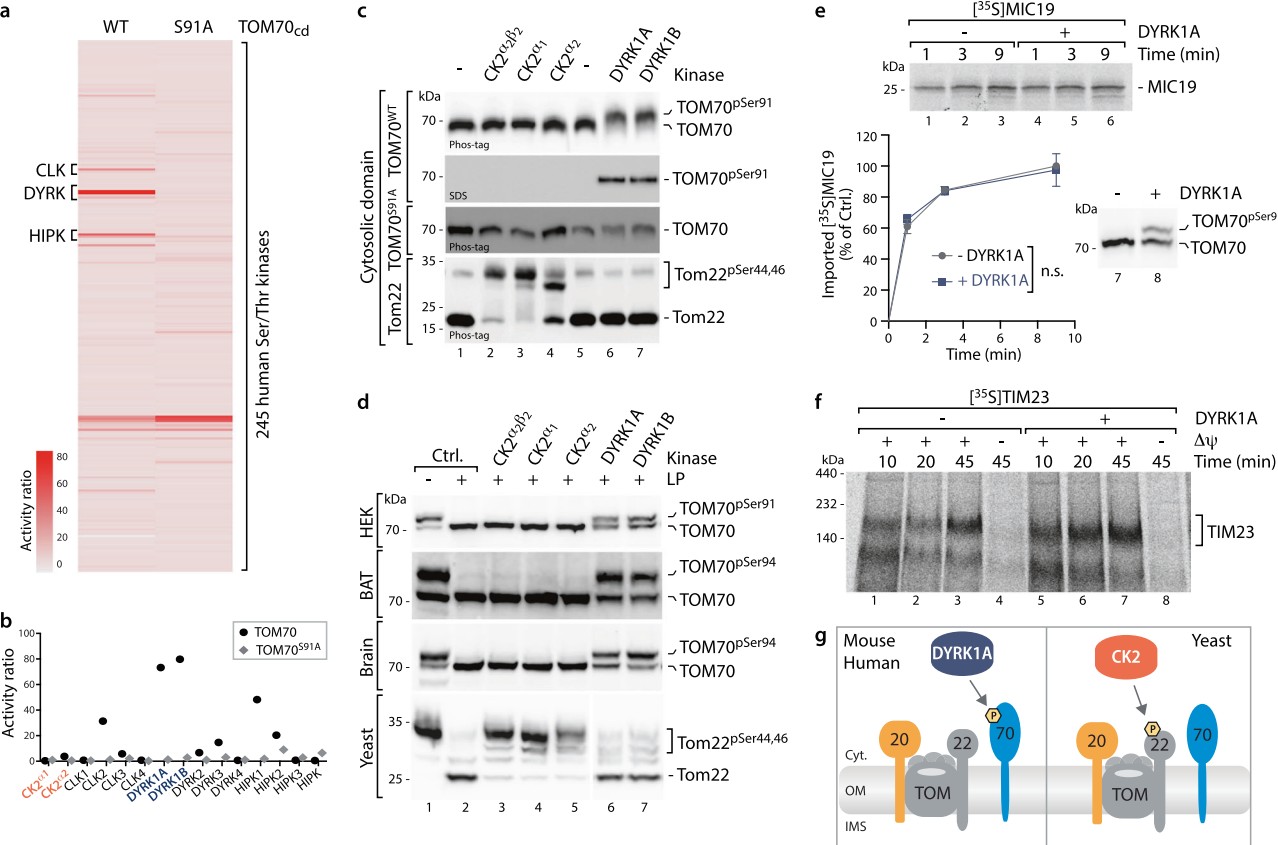

**Fig. 2 | TOM70 is phosphorylated by DYRK1A but not CK2. a** Heatmap of global profiling of TOM70[WT] and TOM70[Ser91Ala] receptor domains for phosphorylation activities of 245 Ser/Thr kinases of the human kinome (KinaseFinder assay[22,36]). HIPK, CLK and DYRK indicate the members of the DYRK-family[74]. **b** Detailed comparison of phosphorylation activities for CK2 isoforms and the DYRK family members for TOM70[WT] and TOM70[Ser91Ala] receptor domains as shown in **a**. **c** In vitro phosphorylation assay of purified receptor domains of human TOM70[WT], TOM70[Ser91Ala] and yeast Tom22 in the presence of indicated kinases. Three different CK2 variants were tested (lanes 2–4). Samples were analyzed via Phos-tag gels and probed with TOM70 or Tom22 antisera or via standard SDS-PAGE and probed with phospho-specific TOM70[pSer91] antibody. **d** Isolated mitochondria from HEK293T cells, yeast, mice brain or brown adipocyte tissue (BAT) were analyzed via Phos-tag gels. For testing phosphorylation with indicated kinases endogenous phosphorylation sites were removed by lambda phosphatase (LP) treatment prior to kinase incubation. Three different CK2 variants were tested (lanes 3–5). **e** Import of [35S]MIC19 precursor protein into LP treated mitochondria after re-phosphorylation of TOM70[Ser91] by DYRK1A (+). (−) mock treated control sample. n.s. not significant. Data were obtained from three biological replicates and represent mean ± SEM. A multiple paired t test with a false discovery rate (FDR) of 1% and a two-stages step-up method of Benjamini, Krieger and Yekutieli was performed to compare between two groups. See the methods section for details on statistical analyses. **f** Import of [35S]TIM23 precursor protein into mitochondria obtained from **e**. Import was monitored via autoradiography after BN-PAGE. **g** Cartoon illustrating targeting of the human, mouse and yeast TOM complex by cytosolic kinases. DYRK1A phosphorylates the human and murine TOM70 receptor at position Ser91 and Ser94, respectively. CK2 targets the yeast Tom22 protein at position Ser44 and Ser46. Source data are provided as a Source Data file.

TOM70[Ser91] phosphorylation (Fig. 2f)[22]. Thus, we conclude that CK2 does not target human TOM70 and does also not play a role in MIC19 import. Instead, TOM70[Ser91] can be phosphorylated by DYRK family kinases and this phosphorylation plays a critical role in activating the carrier import pathway (Fig. 2g)[18,22].

### Human MIA substrates MIC19 and CHCHD6 require TOM20 as import receptor

The proposed inhibitory role of CK2-dependent phosphorylation of TOM70[Ser91] on MIC19 import is the basis for the CK2-TOM70 model[23]. As our molecular analyses revealed that CK2 does not target TOM70 and that TOM70[Ser91] phosphorylation has no effect on MIC19 import (Supplementary Fig. 1c, d; Fig. 2), we wondered whether MIC19 import indeed depends on the TOM70 import receptor. Several studies, from yeast to human, have consistently found MIC19 as a substrate of the MIA40 import pathway, in which the incoming precursor protein is trapped in the intermembrane space by oxidation of a characteristic cysteine motif that is found in human MIC19 (twin CX9C motif) and in yeast Mic19 (non-canonical CX10C motif)[24,38–42]. However, much less is known about the role of the TOM receptors for MIC19 import, which has so far only been studied in yeast: Sakowska et al. found a dependence of Mic19 import on Tom70 and Tom5, but did not test for Tom20[24]. In contrast, Ueda et al. proposed that Tom20 is the import receptor and binds the Mic19 precursor at a conserved myristoyl group[38,39,43].

We set out to systematically investigate which import receptor human MIC19 might require. Since several MIA40 dependent precursor proteins have been found to be imported independently of the major TOM receptors and to require only the TOM import pore[39,44], we first tested whether MIC19 requires any mitochondrial surface receptor. For this, we performed a mild trypsin treatment which removes the critical TOM receptor domains (Fig. 3a)[45]. Import of the classical receptor-dependent presequence import pathway was impaired (shown for the OTC precursor as control (Supplementary Fig. 2a)). MIC19 import was also impaired, indicating the requirement of a surface import receptor (Fig. 3b). To profile which of the major import receptors are required, we imported human MIC19 precursor into mitochondria isolated from cells after siRNA mediated knockdown

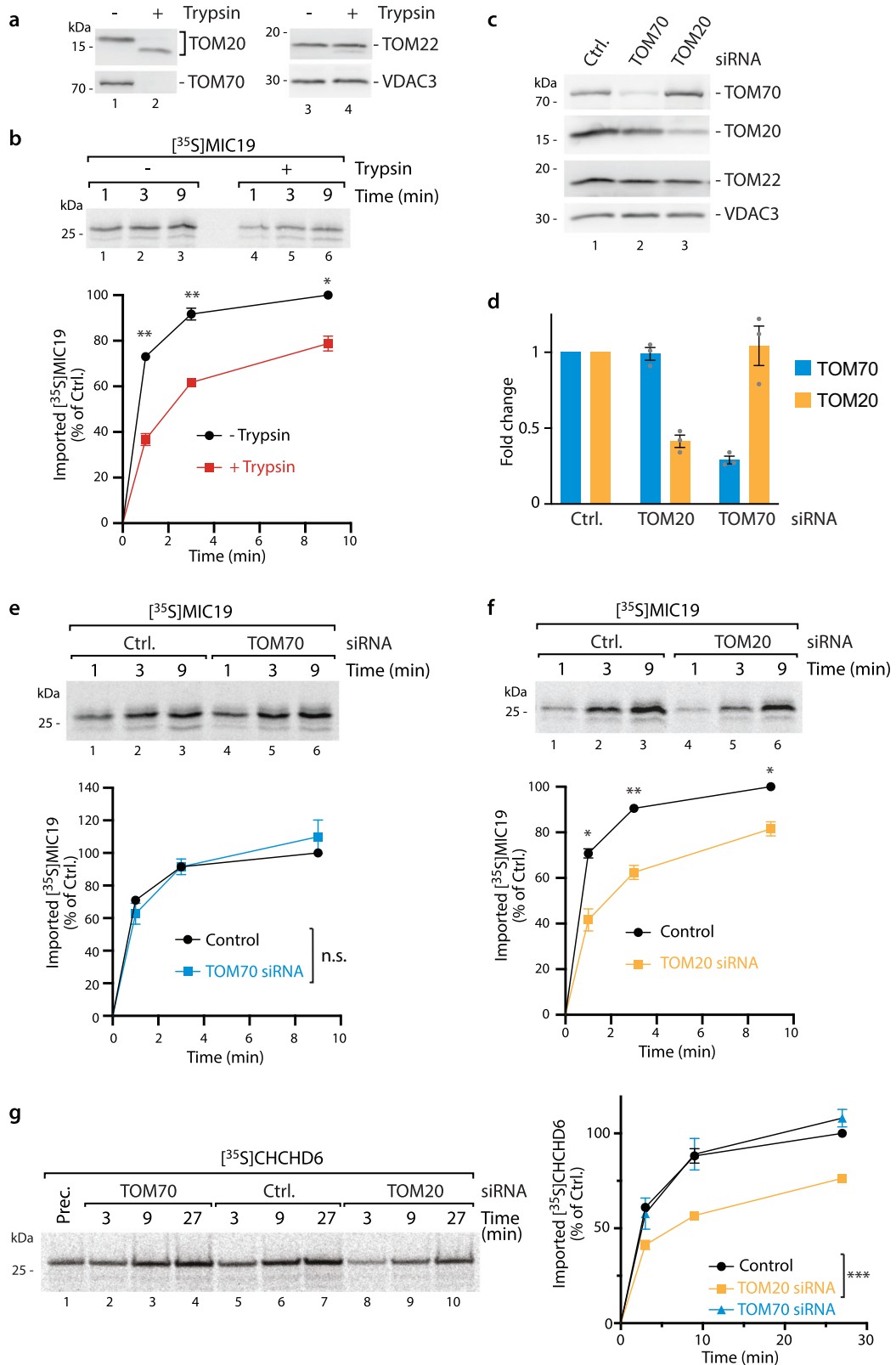

(KD) of TOM70 or TOM20. We tested several time spans and found 48 h optimal to see a specific decrease in the respective import receptors while other import receptors remained unaffected (Fig. 3c, d). Longer knockdown times can easily lead to secondary effects, and thus misinterpretation, as the import of the TOM proteins is also dependent on each other[11,46,47]. To our surprise, MIC19 import was unchanged in TOM70 KD mitochondria (Fig. 3e). The classical

TOM70-dependent TIM23 import was tested as control and reduced as expected (Supplementary Fig. 2b). Remarkably, mitochondria from TOM20 siRNA cells displayed a clear reduction of MIC19 import (Fig. 3f), similar to the classical TOM20-dependent OTC import (Supplementary Fig. 2c)[48]. Thus, we conclude that MIC19 import is independent on TOM70 but instead requires TOM20 as import receptor. We wondered whether other MIA substrates also depend on TOM20

**Fig. 3 | In organello import of human MIA substrates MIC19 and CHCHD6 depend on TOM20 but not TOM70. a** Analysis of mitochondria that were incubated with (+) or without (−) Trypsin after SDS-PAGE and Immunoblotting. **b** Import of [35S]MIC19 precursor protein into mitochondria obtained from **a**. For quantification the control reaction (without Trypsin) at 9 min import time point was set to 100%. Data represent mean ± SEM from three independent experiments. A multiple paired t test with a false discovery rate (FDR) of 1% and a two-stages step-up method of Benjamini, Krieger and Yekutieli was performed to compare between two groups. See the methods section for details on statistical analyses. **\*\****p* < 0.01, and **\****p* < 0.05, n.s. not significant, *p* > 0.05. **c** Immunoblots of isolated mitochondria after depletion of indicated TOM receptors via siRNA. Ctrl., non-targeting siRNA. **d** Quantification of TOM20 and TOM70 knockdown efficiency analyzed in **c**. Data

represent mean ± SEM from three independent experiments. **e** Import of [35S]MIC19 precursor protein into TOM70 depleted and control (Ctrl.) mitochondria. Quantification was performed as in **b**. Data represent mean ± SEM from three independent experiments. For statistical analyses see b and methods section. **f** Import of [35S]MIC19 precursor protein into TOM20 depleted and control (Ctrl.) mitochondria. Quantification was performed as in **b**. Data represent mean ± SEM from three independent experiments. For statistical analyses see b and methods section. **g** Import of [35S]CHCHD6 precursor protein into TOM20 and TOM70 depleted and control (Ctrl.) mitochondria. Quantification was performed as in **b**. Data represent mean ± SEM from four independent experiments. For statistical analyses see b and methods section. Source data are provided as a Source Data file.

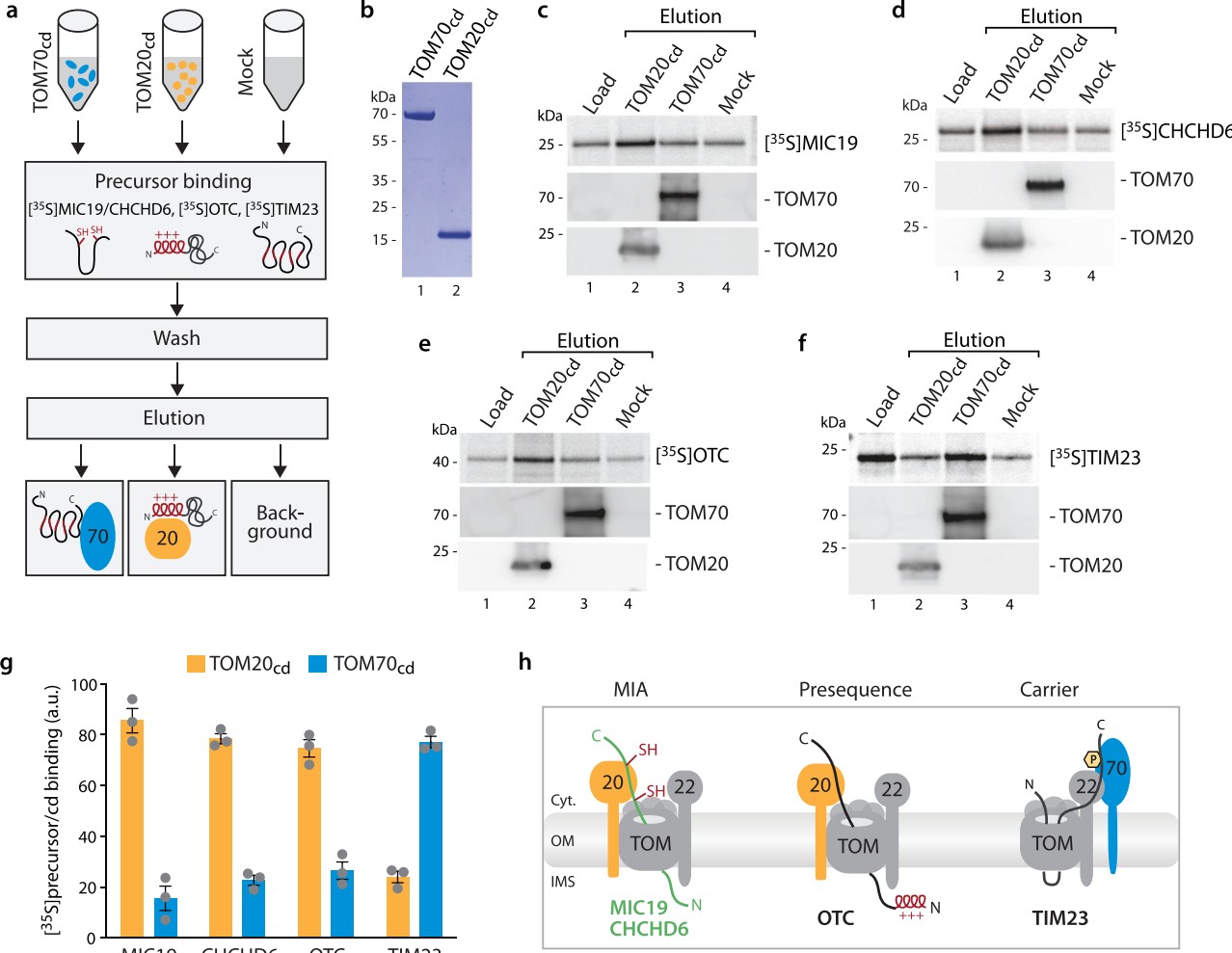

**Fig. 4 | Human MIC19 and CHCHD6 precursor bind to TOM20 receptor domain. a** Schematic overview of the precursor-receptor binding assay. **b** Coomassie staining of SDS-PAGE showing the purified receptor domains of human import receptors TOM20 and TOM70. cd, cytosolic domain. **c** Binding assay for [35S]MIC19 precursor (upper panel) to TOM70cd and TOM20cd (immobilized via decaHis-tag on Ni·NTA beads). Load, loaded radiolabelled precursor; Elution, elution fraction containing released TOM70cd or TOM20cd and radiolabelled precursor in case of specific binding. Mock, same treatment without TOM receptor domains detects

non-specific binding. **d** Binding assay for [35S]CHCHD6 precursor performed as in **a**. **e** Binding assay for [35S]OTC precursor performed as in **a**. **f** Binding assay for [35S]TIM23 precursor performed as in **a**. **g** Quantification of specific precursor binding to TOM20cd or TOM70cd, respectively. Quantification from three independent experiments ± SEM. a.u., arbitrary units. **h** Cartoon illustrating MIC19 precursor import via TOM20- and TIM23 carrier import via TOM70-associated TOM core complexes. Source data are provided as a Source Data file.

and tested CHCHD6 that like MIC19 harbors the canonical twin CX9C motif of MIA substrates[49]. Indeed, we find that also CHCHD6 import is delayed in TOM20 KD mitochondria, while import efficiency upon TOM70 depletion was similar to control mitochondria (Fig. 3g). Thus, import of both tested human MIA substrates, MIC19 and CHCHD6, seem to depend on TOM20 but not on TOM70.

To further confirm this unexpected finding, we included two additional experimental approaches, that are both independent on the manipulation of TOM70 and TOM20 expression via siRNA treatment: (i) we tested direct binding of MIC19 and CHCHD6 precursors to purified human TOM20 and TOM70 receptor domains (cd) (Fig. 4a, b)[22,50] and found a clear preference for their binding to

$TOM20_{cd}$, like the established TOM20 substrate OTC (Fig. 4c–e, g). In contrast, $TOM70_{cd}$ was not binding MIC19 or CHCHD6, whereas the carrier model precursor TIM23 showed clear $TOM70_{cd}$ binding (Fig. 4f, g). (ii) We tested if the presence of cytosolic receptor domains of TOM20 and TOM70, respectively, can compete with MIC19 import in organello[18]. We found that MIC19 import is delayed in the presence of $TOM20_{cd}$ but not $TOM70_{cd}$ (Supplementary Fig. 2d–g), confirming requirement for TOM20 but not TOM70.

Taken together, using independent experimental approaches, we show here that the import of the human MIA substrates MIC19 and CHCHD6 is dependent on the import receptor TOM20, like the classical presequence precursor OTC, but does not require TOM70 (Fig. 4h). Furthermore, our results do not support the model proposed by Latorre-Muro et al.[23] that MIC19 import and consequently cristae biogenesis is regulated via a CK2-TOM70 signaling axis.

## CK2 inhibitor CX4945 activates TOM20-dependent import via off-target inhibition of DYRK1A

Our results that (i) MIC19 import and TOM70 phosphorylation are not affected by β-adrenergic stimulation in BAT cells, (ii) DYRK1A but not CK2 targets TOM70, (iii) MIC19 import is not dependent on $TOM70^{Ser91}$ phosphorylation and that (iv) TOM20 but not TOM70 is the import receptor for MIC19, clearly strengthen the model of the DYRK1A-TOM70 axis in shaping of the mitochondrial proteome. However, we still wondered whether we could find an experimental explanation for the results that led to the proposal of a CK2-TOM70 axis[23]. A key experimental tool from which the model was deduced is the CK2 inhibitor CX4945, which was suggested to increase MIC19 biogenesis by impairing the inhibitory CK2-dependent TOM70 phosphorylation. CX4945, also known as Silmitasertib, is used in clinical trial studies for treatment of various cancers and COVID-19[51–53]. Reviewing the literature, we found that CX4945 is also a potent inhibitor of DYRK1A[54–56]. Intrigued by these reports, we speculated that such CX4945 off-target inhibition might affect DYRK1A-dependent phosphorylation of $TOM70^{Ser91}$. Inhibition of DYRK1A using the highly selective inhibitor INDY drives a transcriptional response to compensate for the loss of TOM70 phosphorylation. This encompasses a strong upregulation of DYRK1A and DYRK1B, but also of all TOM import receptors, including TOM20[22]. We therefore hypothesized that off-target inhibition of DYRK1A by the CK2 inhibitor CX4945 could lead to a similar response and result in upregulation of the TOM20 import receptor. This in turn would likely stimulate the TOM20-dependent import of MIC19. Such feedback loop may ultimately explain an activating role of CX4945 in MIC19 biogenesis in vivo, however, this mechanism is completely detached from a CK2-TOM70 axis[23].

To pursue this idea, we first analyzed if CX4945 can inhibit the established DYRK1A-dependent phosphorylation of TOM70[22]. We tested this with purified TOM70 receptor domain in vitro and with isolated mitochondria in organello. In both approaches, CX4945 efficiently inhibited TOM70 phosphorylation by DYRK1A, similar to the specific DYRK1A inhibitor INDY (Fig. 5a, b). In parallel, we confirmed the concomitant inhibition of CK2 by CX4945 using yeast Tom22 as established CK2 target[18] (Fig. 5c). Next, we analyzed if application of CX4945 in cell culture could stimulate the expression of TOM receptors similarly to the DYRK1A specific inhibitor INDY[22]. Indeed, we found a stimulation of the expression of all three TOM receptors as well as DYRK1A and DYRK1B (Fig. 5d). Transcripts of the import pore TOM40 and the control VDAC3 were not changed. Intriguingly, *TOMM20* showed the highest stimulation of the three TOM import receptors, which was even higher with CX4945 than with INDY. To assess if this altered transcription rates ultimately also affect protein levels, we performed immunoblot analysis and confirmed the up-regulation of all three import receptors by CX4945 (Fig. 5e, f; TOM40 and GRP75 levels were largely unaffected). Finally, we isolated mitochondria from cells cultured in the presence of the CK2 inhibitor CX4945 and tested for

MIC19 import. Indeed, we found a stimulation of MIC19 import when CX4945 was applied in vivo which is likely caused by increased TOM20 protein levels (Fig. 5g).

Taken together, our findings provide molecular insight into the interplay of cytosolic signaling and mitochondrial protein biogenesis via the DYRK1A-TOM70 axis and an alternative explanation for the observed increased MIC19 biogenesis upon CX4945 addition in vivo: CX4945, through off-target inhibition of DYRK1A, leads to a transcriptional stimulation of the actual MIC19 import receptor TOM20 and thus enables stimulation of MIC19 import (Fig. 5h).

## Impaired DYRK1A signaling activates the presequence import pathway

Intrigued by the uncovered signaling crosstalk at the DYRK1A-TOM70 axis and its stimulatory role of the MIA import pathway, we speculated that remodeling of the import machinery upon DYRK1A inhibition may also stimulate the classical presequence import pathway which is primarily dependent on TOM20. This pathway represents the largest import route into mitochondria as approx. 70% of all precursor proteins harbor N-terminal presequences[40]. These precursors are directed after passage across the outer membrane to the TIM23 complex in the inner membrane where they are translocated in a membrane potential dependent manner into the matrix followed by cleavage of the presequence by the matrix processing protease MPP[40]. We tested the human mitochondrial HSP60 precursor HSPD1 as a model substrate and analyzed import kinetics into mitochondria isolated from cells that had been either treated with the specific DYRK1A inhibitor INDY or the CK2 inhibitor CX4945, that also inhibits DYRK1A via an off-target effect (Fig. 5a, b). Indeed, both inhibitors caused a clear stimulation of HSPD1 import compared to mock treated control mitochondria (Fig. 6a, b). Thus, inhibition of DYRK1A and its concomitant remodeling of the import machinery does not only stimulate MIC19 import via the MIA pathway but also enhances HSPD1 import via the presequence import pathway. Therefore, DYRK1A emerges as critical regulatory hub that connects and fine-tunes the central mitochondrial import pathways (Fig. 6c).

## Discussion

The identification of the first human protein kinases that target the mitochondrial import machinery by Walter et al.[22] and Latorre-Muro et al.[23] has attracted considerable attention and revealed a new level of complexity in rewiring of the mammalian metabolism by cellular signaling networks that directly act on the organellar entry gate. However, while both studies identified the same TOM70 import receptor and the same conserved serine residue as target, their suggested kinases (DYRK1A vs. CK2) as well as their proposed regulatory roles of this site (activation of the metabolite carrier import pathway vs. inhibition of the import of the single intermembrane space protein MIC19) were not reconcilable. To resolve this controversy, we systematically dissected and experimentally studied both TOM70 phosphorylation models in molecular detail[22,23]. Our results show that TOM70 phosphorylation levels and MIC19 import do not change in differentiated BAT cells upon NE induction. Furthermore, using several independent approaches, we show that CK2 does not target TOM70 and that TOM70 is not the import receptor of MIC19. In contrast, we identify TOM20 as the import receptor of human MIC19 and provide experimental data confirming DYRK1A as the kinase that targets TOM70 to stimulate the carrier import pathway.

How can these results and the findings in the initial study[23] be explained? For the identification of CK2 as mammalian TOM70 kinase, Latorre-Muro et al.[23] refer to two publications: (i) Schmidt et al.[18] discovered CK2 as kinase of the central import receptor Tom22 in *S. cerevisiae*, in which CK2-dependent phosphorylation plays a stimulatory role in protein import[18]. (ii) Shinoda et al. identified mouse

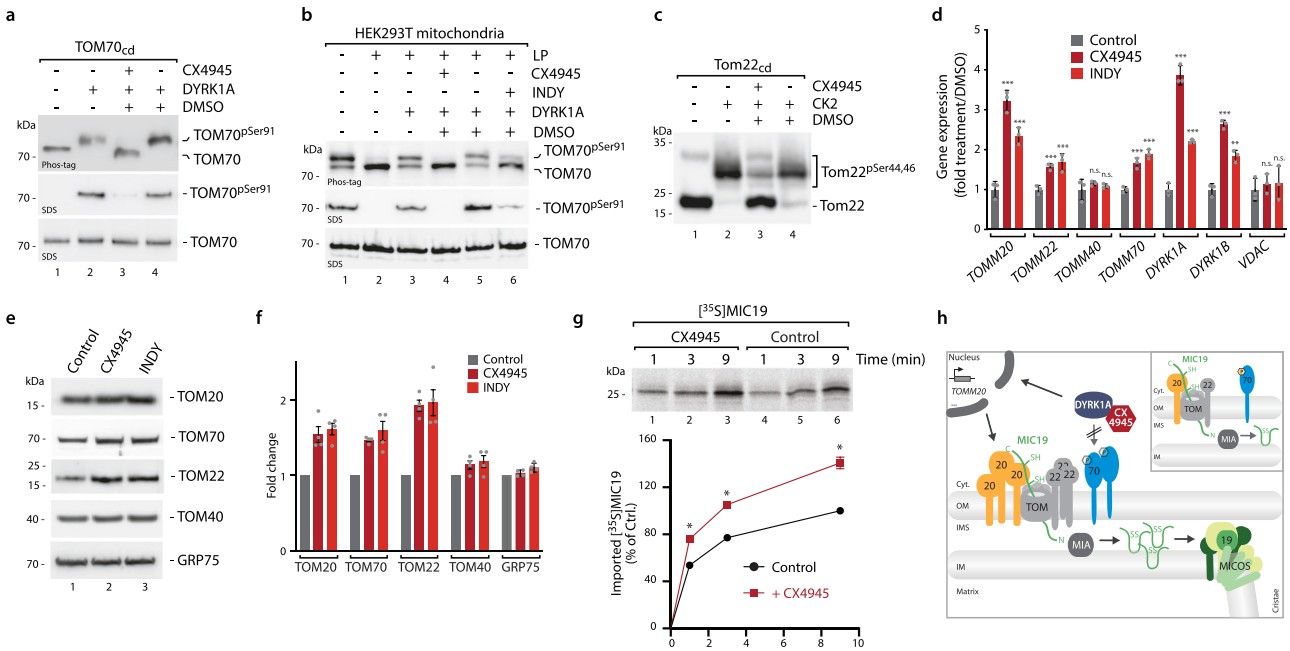

**Fig. 5 | CK2 inhibitor CX4945 inhibits DYRK1A leading to a transcriptional response upregulating the mitochondrial import receptors. a** Inhibition of DYRK1A dependent phosphorylation of TOM70 receptor domain in the presence of the CK2 inhibitor CX4945. Upper panel, Phos-tag gel; Middle and lower panel, SDS-PAGE. **b** Inhibition of DYRK1A-dependent phosphorylation of mitochondrial TOM70 by CX4945 and INDY. Endogenous phosphorylation was removed by lambda phosphatase (LP) before DYRK1A incubation. **c** Phos-tag gel showing inhibition of CK2-dependent phosphorylation of yeast Tom22 by CX4945. **d** Analysis of changes of indicated transcript levels upon INDY and CX4945 treatment (12 h) by qRT-PCR. $n = 3$, data represent mean ± SEM. Statistical analysis was performed using a two-sided Student's $t$ test to compare between two groups (***$p < 0.001$; **$p < 0.01$; n.s. not significant.) Data are representative of three independent experiments. **e** Analysis of indicated protein levels after in vivo treatment with CX4945 or INDY (12 h) by SDS-PAGE and immunoblotting. **f** Quantification of indicated proteins from **e**. Statistical analysis was performed using a two-sided Student's $t$ test to compare between two groups (Control vs. CX4945 or Control vs.

INDY). Data represent mean ± SEM from four independent experiments. *$p < 0.05$. **g** Import of [35S]MIC19 precursor protein into mitochondria from cells incubated in the presence or absence of CX4945 for 12 h. For quantification the control reaction at 9 min import time point was set to 100%. A multiple paired t test with a false discovery rate (FDR) of 1% and a two-stages step-up method of Benjamini, Krieger and Yekutieli was performed to compare between two groups. See the methods section for details on statistical analyses. Data represent mean ± SEM from three independent experiments. **h** Model for stimulation of MIC19 import via CX4945: Import of human MIC19 requires the import receptor TOM20 but is independent of TOM70 (inset). Inhibition of DYRK1A which normally phosphorylates TOM70 for activation of the carrier import pathway triggers a transcriptional response stimulating expression of TOM receptors including the MIC19 receptor TOM20. This stimulates the TOM20-dependent import pathway. Imported MIC19 precursor become trapped in the intermembrane space by oxidation at the MIA machinery. This may then stimulate MICOS biogenesis and ultimately control of cristae structure and respiration. Source data are provided as a Source Data file.

TOM70^Ser94 as target of CK2 in beige adipocytes[57]. In this mass spectrometry-based high throughput study a cell extract was treated with a thermosensitive phosphatase, followed by thermal phosphatase inactivation and incubation with recombinant CK2. All subsequently identified phosphorylation sites were considered CK2 targets. Notably, this qualitative study did not analyze phosphorylation prior to CK2 treatment, and thus abundant sites, such as TOM70^Ser94, could easily be detected (and counted as a CK2 target) when phosphatase treatment would not be 100% efficient. Due to this limitation, the co-senior author published a follow-up study with quantitative profiling for substrates (including comparison with non-kinase treatment) in which TOM70 was no longer identified as a CK2 target[57]. As this published literature does not support phosphorylation of TOM70 by CK2 and as such experimental data has also not been provided in the study proposing the TOM70-CK2 axis[23], we investigated the potential of CK2 to phosphorylate TOM70 by several independent experimental approaches. However, we find no experimental evidence that CK2 targets TOM70 and therefore no support for the existence of a CK2-TOM70 axis.

We next addressed the question which TOM receptor MIC19 requires for its mitochondrial import. For this we applied siRNA mediated protein knockdown and found that MIC19 import decreases only upon depletion of TOM20, but not TOM70. We validated this finding by additional experiments including in vitro binding assays and

in organello competition experiments, both supporting MIC19 dependency on TOM20. Additional data show that TOM20 is also the import receptor for the CHCHD6 precursor that like MIC19 belongs to the family of MIA substrates with the typical twin CX₉X motifs[49]. It is therefore likely that the involvement of TOM20 as import receptor of the MIA import pathway in human mitochondria represents a common principle. However, this is in contrast to the TOM70-CK2 model that also used protein depletion and concluded that TOM70 is the import receptor for MIC19[23]. A possible explanation for this discrepancy is that severe TOM70 depletion resulted in secondary effects and a decrease in additional TOM subunits (controls actually indicated reduction of several mitochondrial inner membrane proteins, whereas TOM20 levels were not assessed). This scenario is highly likely as the TOM receptors depend on each other for their own biogenesis. We therefore speculate that the impaired MIC19 import in TOM70 depleted mitochondria observed in Latorre-Muro et al. is caused by a co-depleted TOM20 receptor or co-depleted components of the MIA import machinery.

As our detailed experimental testing reveal that CK2 does not target TOM70 and that MIC19 requires TOM20 but not TOM70 as import receptor, the proposed CK2-TOM70 axis is unsubstantiated and with this, the deduced model of increased MIC19 import and its role in enhancing cristae formation and respiration upon cold stress and β-adrenergic responses has to be critically challenged[31,58–62].

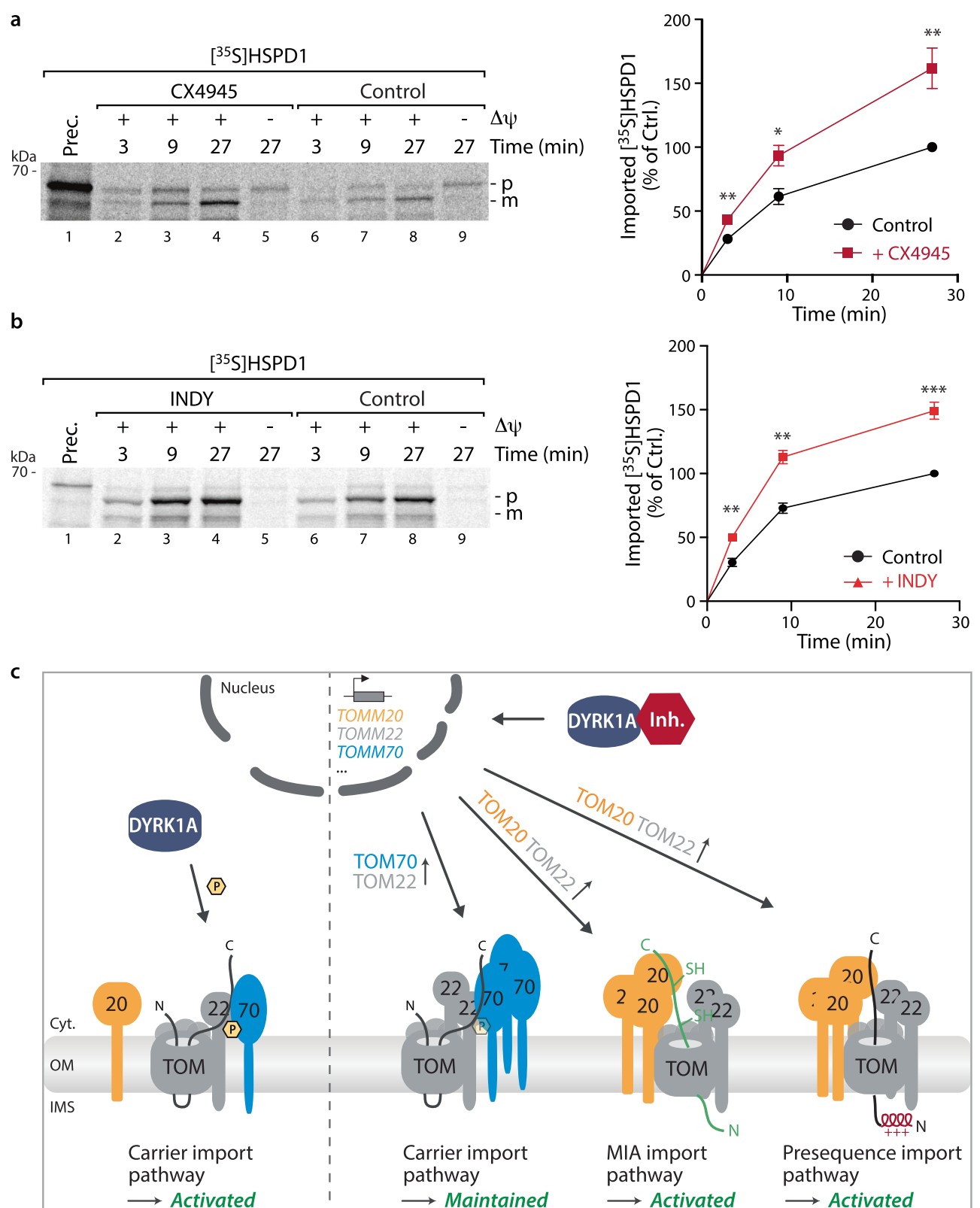

Interestingly, Latorre-Muro et al. described increased MIC19 import also after in vivo treatment with the CK2 inhibitor CX4945. Intrigued by this finding, we dissected the molecular mechanism behind this observation and revealed that the increased MIC19 import is in effect mediated by off-target inhibition of DYRK1A by CX4945. This in turn leads to up-regulation of import receptors, including TOM20, and consequently to enhanced MIC19 import.

Moreover, we find that upregulation of the TOM import receptors upon DYRK1A inhibition also activates the classical presequence import pathway. This pathway is relevant for approx. 70% of all precursors destined to mitochondria[40] and depends largely on the import receptor TOM20. We observe activation of this positive feedback loop with both inhibitors, the specific DYRK1A inhibitor INDY as well as the CK2 inhibitor CX4945, however, here the effect is

**Fig. 6 | DYRK1A signaling links the main mitochondrial import pathways.**
**a** Import of [$^{35}$S]HSPD1 precursor protein into mitochondria from cells incubated in the presence or absence of the CK2 inhibitor CX4945 for 12 h. p precursor, m mature. For quantification the control reaction at 9 min import time point was set to 100%. Data represent mean ± SEM from three independent experiments. A multiple paired t test with a false discovery rate (FDR) of 1% and a two-stages step-up method of Benjamini, Krieger and Yekutieli was performed to compare between two groups. See the methods section for details on statistical analyses. *$p < 0.05$; **$p < 0.01$. **b** Import of [$^{35}$S]HSPD1 precursor protein into mitochondria from cells

incubated in the presence or absence of the DYRK1A inhibitor INDY for 12 h. Quantification performed as in **a**. ***$p < 0.001$. **c** Schematic showing the role of DYRK1A as central regulatory hub linking the main mitochondrial import routes. While DYRK1A is essential to activate the carrier import pathway via TOM70$^{pSer91}$ phosphorylation (left panel), its impairment leads to a transcriptional activation of all three TOM import receptors followed by a remodeling of the TOM complex that allows an activation of the MIA- and the presequence import pathways (right panel). Source data are provided as a Source Data file.

due to its off-target inhibition of DYRK1A. Thus, the DYRK1A-TOM70 axis emerges as central regulatory hub that links the main mitochondrial protein import pathways.

Our finding that CX4945 modulates mitochondrial protein biogenesis via the DYRK1A-TOM70 axis activating import of MIA and also of presequence precursor substrates might also have clinical implications, as the inhibitor is used in clinical trials for the treatment of various cancers and COVID-19[51–53]. Interestingly, CX4945 has recently been found to have similar effects on mitochondrial respiration as the DYRK1A inhibitor INDY[22,63]. Moreover, the feedback loop linking DYRK1A inhibition to modulation of the main mitochondrial protein biogenesis pathways may also help to elucidate the molecular mechanisms underlying the clinical manifestations of *DYRK1A syndrome*: Patients with dysfunctional *DYRK1A* present with symptoms such as motor deficits, microcephaly, speech delay, intellectual disability and autism, which, especially in their combinations, resemble typical mitochondrial disease phenotypes[22,64–69]. It remains to be determined whether loss of function mutations of *DYRK1A*, similar to chemical inhibition of DYRK1A, lead to a remodeling of the mitochondrial import machinery. Notably, DYRK1A is also encoded on chromosome 21 and expressed 1.5 fold in Down syndrome patients. It will also be interesting to test whether increased levels of DYRK1A might interfere with the feedback loop and thus impact on mitochondrial protein biogenesis. The discovery of the DYRK1A-TOM70 axis as regulatory hub that links the main mitochondrial protein import pathways now paves the way to reach a mechanistic understanding of the described links of Down syndrome and dysfunctional mitochondrial energy metabolism[70,71].

Taken together, using detailed molecular analysis our study resolves the discrepancies that emerged from the two different TOM70 phosphorylation models and uncovers the DYRK1A-TOM70 axis as regulatory hub which links the major mitochondrial protein import pathways. Thus, DYRK1A emerges as surveillance kinase for mitochondrial protein biogenesis as modulation of DYRK1A activity adapts the central mitochondrial protein entry gate for rewiring of the mitochondrial metabolism.

## Methods

### Animals and tissue culture
As experimental model systems we used human embryonic kidney cells (HEK293T), differentiated primary immortalized brown adipocytes (PIBA) cells and mouse brain or brown adipose tissue (BAT) from C57BL6/N mice. Mice were bred and kept in the animal facilities of the University Medical Center Freiburg according to institutional guidelines and the German law for animal protection. Mice were maintained in specific pathogen-free conditions, fed with standard diet and had access to food and water *ad libitum*. Climate conditions in the animal facility were set to 21–23 °C and a 45–60% humidity. A day/night rhythm of 12 h light and 12 h darkness was maintained. Sampling of tissue from sacrificed mice was approved by the government commission for animal protection and the local ethics committee (X-18/10C). Eight weeks old male mice (for brain tissue) and 6 weeks old mice (for BAT) were euthanized by cervical dislocation and respective tissues removed.

HEK293T cells were cultured in DMEM containing 10% (v/v) fetal bovine serum (FBS), 2 mM L-glutamine and 4.5 g/l glucose. Cells were grown at 37 °C in a humidified incubator with 5% $CO_2$. Differentiation of PIBA cells was performed as described previously[32]. In short, differentiation was initiated using the growth medium described above supplemented with 1 nM T3, 20 nM Insulin, 0.5 μM Dexmethasone, 0.125 mM Indomethacin and 0.5 μM Isobutylmethylxanthine (IBMX). After 24 h cells were differentiated using medium supplemented with 1 nM T3 and 20 nM Insulin. After 4 days the growth medium was supplemented with 1 μM Norepinephrine and incubated for 18 h. Differentiation was monitored by Oil Red O (ORO) staining[32]. Knockdowns were generated by reverse transfection of HEK293T cells with a final concentration of 33 nM of the respective siRNA oligonucleotides using Lipofectamine™ RNAiMAX. Non-targeting siRNA (Eurogentec, #SR-CL000-005) was used as a negative control. Experiments were started 48 h after transfection. For TOM20 knockdown the sequence 5′GAUGCUGAAGCUGUUCAGA3′ and for TOM70 knockdown 5′GAAGGAAUGUUUAGAAGAU3′ were used.

### Isolation of mitochondria
Mitochondria were isolated as previously described by Johnston et al.[72]. Briefly, tissue and cell pellets were homogenized in solution A (220 mM mannitol, 70 mM sucrose, 20 mM HEPES-KOH, pH 7.6, 1 mM EDTA, 0.5 mM PMSF and 2 mg/ml BSA) using a glass potter. Cellular debris was removed by centrifugation (5 min at $800 \times g$ and 4 °C) and mitochondria were isolated from the recovered supernatant by centrifugation at $10,000 \times g$ (15 min at 4 °C). Mitochondrial pellet was then resuspended in solution B (solution A without BSA) and again centrifuged at $800 \times g$ for 5 min (4 °C). Mitochondria were pelleted by a $10,000 \times g$ spin for 15 min and resuspended in import buffer (described below)[22]. Protein concentrations were determined by Bradford assay and samples stored at −80 °C or directly used for functional assays.

### Purification of TOM receptor domains
Cytosolic domains of human TOM70 (residues 60–608), human TOM20 (35–145) and yeast Tom22(1–97) were expressed and purified via N-terminal deca Histidine tags as described[18,22]. For the TOM70$_{cd}$$^{S91A}$ variant, the point mutation was introduced by site-directed mutagenesis[22]. Briefly, *E. coli* cells were transformed with the respective expression plasmids and grown at 37 °C until an $OD_{600}$ of 0.6. After addition of 1 mM IPTG and incubation for 3 h cells were harvested and resuspended in extraction buffer (4 mM $MgCl_2$, 0.5 mM PMSF, 20 mM Tris/HCl, pH 8.0). 1 mg/ml lysozyme and DNase I were added and after incubation for 30 min on ice lysates were cleared at $10,000 \times g$ for 10 min at 4 °C. Supernatants were incubated with Ni-NTA resin beads (Qiagen) and washed three times with washing buffer (20 mM Tris/HCl, pH 8.0, 50 mM Imidazole, 200 mM NaCl, 1 mM β-mercaptoethanol). Proteins were eluted with elution buffer (20 mM Tris/HCl, pH 8.0, 350 mM Imidazole, 100 mM NaCl, 1 mM β-mercaptoethanol). Protein purity was analyzed by SDS-PAGE and Coomassie brilliant blue staining.

## In vitro and in organello phosphorylation

For phosphorylation of the purified receptor domains 2 µg protein was incubated in 30 µl ST buffer (250 mM sucrose, 10 mM $MgCl_2$, 1 mM PMSF, 10 mM Tris/HCl, pH 7.2) supplemented with 5 mM ATP and 1 µl of the respective protein kinase (CK2$^{\alpha2\beta2}$, NEB, Cat#P6010S; CK2$^{\alpha1}$, ThermoFisher Scientific, Cat#PV3248; CK2$^{\alpha2}$, ThermoFisher Scientific, Cat#PV3624; DYRK1A, ThermoFisher Scientific, Cat#PR7189B or DYRK1B, ThermoFisher Scientific, Cat#PR8350A). After incubation for 45 min at 30 °C and gentle agitation at 500 rpm Laemmli buffer was added to stop the reaction. Phosphorylation was monitored via Phos-tag gel electrophoresis and immunoblotting (see below)[18,22,37]. For in organello phosphorylation, 50 µg isolated mitochondria were incubated in 50 µl kinase assay buffer (10 mM Tris/HCl, pH 7.4, 250 mM sucrose, 10 mM $MgCl_2$, 1 mM PMSF, 1× PhosStop (Roche), 1× *EDTA free protease inhibitor cocktail* (Roche) and 1× *Protein Metallo Phosphatase* (PMP) buffer (NEB)). For dephosphorylation 1 µl lambda phosphatase (NEB) was added and sample was incubated for 30 min at 37 °C with gentle agitation at 550 rpm[22]. Mitochondria were washed twice with sucrose buffer (0.5 M sucrose, 10 mM HEPES/HCl, pH 7.6) and resuspended in assay buffer containing 5 mM ATP and 1 µl of the respective protein kinase (with or without CK2 inhibitor CX4945 (10 µM final concentration; Selleck Chemicals, Cat#S2248)[23]. After incubation at 37 °C and gentle agitation at 550 rpm mitochondria were washed with sucrose buffer and resuspended in Laemmli buffer. Phosphorylation was monitored via Phos-tag gel electrophoresis and immunoblotting[22].

## Import of precursor proteins into isolated mitochondria

Radiolabelled human MIC19, CHCHD6, HSPD1, TIM23 and OTC precursor proteins were generated by in vitro transcription/translation in the presence of $^{35}$S-methionine using the rabbit reticulocyte lysate system (Promega)[19,22,40,73]. Isolated mitochondria were incubated with precursor protein in import buffer (250 mM sucrose, 5 mM magnesium acetate, 80 mM potassium acetate, 10 mM sodium succinate, 20 mM HEPES-KOH, pH 7.4) supplemented with 1 mM DTT and 5 mM ATP and incubated for the indicated time points at 37 °C. Proteinase K (20 µg/ml) was added to remove non-imported protein followed by inactivation with PMSF (50 µg/ml for 10 min on ice). Mitochondria were re-isolated by centrifugation at 10,000 × *g* for 10 min at 4 °C and import was analyzed by SDS-PAGE and autoradiography[40].

When indicated isolated mitochondria were pretreated with trypsin (25 µg/ml) for 15 min on ice to remove cytosolic domains of OM proteins. Trypsin treatment was stopped by adding a 30× access of trypsin inhibitor from soy bean (750 µg/ml) for 10 min on ice. Mitochondria were reisolated and used for import. For import of TIM23 mitochondria were lysed after import reaction in Digitonin buffer (1% (w/v) digitonin, 0.5 mM EDTA, 10% glycerol, 50 mM NaCl, 20 mM Tris-HCl, pH 7.4) and subjected to Blue Native electrophoresis followed by autoradiography as described[22]. For dissipation of the membrane potential Δψ, antimycin A (8 µM), valinomycin (1 µM), and oligomycin (20 µM) were added to mitochondria prior to the import reaction. For import competition assays the reticulocyte lysates with radiolabelled precursor proteins were incubated with 1 µg purified receptor domains and then applied for the import reaction as described above.

## Monitoring phosphorylation of human/mouse TOM70 and yeast Tom22

For analysis of phosphorylated proteins $Zn^{2+}$-Phos-tag Bis-Tris gel electrophoresis was performed as described previously[22,37]. For analysis of human and mouse TOM70 phosphorylation, a discontinuous 8% SDS gel was prepared and 12.5 µM Phos-tag reagent (FUJIFILM Wako Chemicals) and 25 µM $ZnCl_2$ were added to the separation gel mix before polymerization (for analysis of yeast Tom22 a 12.5% SDS gel was prepared and 50 µM Phos-tag and 100 µM $ZnCl_2$ were added, respectively). Electrophoresis was performed for 3-4 h at 35 mA and 600 V. Proteins were transferred via wet blotting (Biorad system) onto a PVDF membrane. Protein phosphorylation was monitored via immunodecoration.

Alternatively, TOM70 phosphorylation was monitored with a phosphospecific antibody raised against a phosphorylated peptide corresponding to human TOM70 residues 86-97. The antibody recognizes phosphorylated human TOM70$^{pSer91}$ and the homologous position of mouse TOM70$^{pSer94,22}$.

## Precursor binding to TOM import receptor domains

Human TOM70 and TOM20 receptor domains containing N-terminal deca-His tags were resuspended with $Ni^{2+}$-NTA resin equilibrated with assay buffer (20 mM Imidazole, 5–10 mM KCl, 10 mM MOPS/KOH, pH 7.2, 1% (w/v) BSA). Samples were incubated for 30 min (4 °C) and then transferred to 1 ml Mobicol mini columns. [$^{35}$S]-labeled precursor proteins were diluted in assay buffer (up to max. 5% (v/v) rabbit reticulocyte lysate was used) and added to the resuspended resin. After incubation for 30 min at 30 °C resin was pelleted and washed three times with assay buffer without BSA. Bound proteins were eluted with Laemmli buffer and analyzed by SDS-PAGE, autoradiography, and immunoblotting[22]. For quantification precursor binding to the mock sample (containing no cytosolic protein domain) was substracted from binding to TOM70$_{cd}$ and TOM20$_{cd}$. Obtained total binding for each precursor to TOM70$_{cd}$ and TOM20$_{cd}$ was added and set to 1. Calculation of the precursor to cd binding ratio was used to enable comparison of biological replicates.

## Profiling transcriptional changes upon INDY and CX4945 treatment by qRT-PCR

Cells were incubated in the presence of 10 µM INDY or CX4945 (both taken from a DMSO stock) for 12 h. As control, identical volume of DMSO without inhibitor was applied. Total RNA was isolated using the RNeasy Mini Kit (Qiagen) and treated with DNase to omit DNA contaminations. cDNA was synthesized with High-Capacity cDNA kit (Applied Biosystems) from 2 µg RNA. PCR amplifications and detections were performed with CFX384 Real-time PCR detection system (Bio-Rad) using SsoAdvanced Universal SYBR Green Supermix (Bio-Rad). For normalization *VDAC3* was used as housekeeping gene. Relative mRNA levels were calculated with the delta-delta Ct method[22].

## Statistics and reproducibility

All experiments were replicated at least three times. Data shown represent means +/− standard error of the mean (SEM). For qRT-PCR analysis a Student's *t* test was applied to compare between two groups. For comparison of in organello import kinetics a multiple paired t test with a false discovery rate (FDR) of 1% and a two-stages step-up method of Benjamini, Krieger and Yekutieli was used. Presence of outliers was assessed with the ROUT method and shown significance values are two-sided. Normal distribution of data was confirmed by Shapiro–Wilk's test and by visually inspecting Q-Q plots. Significances are presented as *p* values with ***$p < 0.001$, **$p < 0.01$, and *$p < 0.05$, n.s., not significant, $p > 0.05$. Statistical analysis was performed using Prism 9 (version 9.5.1).

## Reporting summary

Further information on research design is available in the Nature Portfolio Reporting Summary linked to this article.

# Data availability

All data associated with this study can be found in the paper and the supplementary materials. Source data are provided with this paper. Research materials are available upon request at chris.meisinger@biochemie.uni-freiburg.de. Source data are provided with this paper.

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

## Acknowledgements

We thank Dr. Jan-Wilhelm Kornfeld for the stable BAT cell line and scientific advice, Dr. Jan Riemer for the CHCHD6 cDNA, Dr. Carlotta Peselj and Dr. Sabrina Büttner for scientific discussion and Alexandra Schwierzok for expert technical assistance. Our work was supported by the Deutsche Forschungsgemeinschaft (DFG), under Germany's Excellence Strategy (CIBSS - EXC-2189 - Project ID 390939984, the RTG 2606, the SFB1381 (Project-ID 403222702) (to C.M. and F.N.V.), the RTG 2202 (to C.M.), the Emmy-Noether Programm of the Deutsche Forschungsgemeinschaft (to F.N.V.), the Heisenberg-Professorship BR 3662/5 (to T.B.), and the German Cancer Consortium (DKTK, L627).

## Author contributions

A.M., C.W., T.S., A.N., S.S. and I.D. performed the experiments. A.M., C.W., T.S., T.B., F.N.V. and C.M. designed experiments, analyzed and interpreted the data. F.N.V. and C.M. wrote the manuscript. C.M. coordinated and directed the project. All authors approved the final version of the manuscript.

## Funding

## Competing interests

The authors declare no competing interests
