## [Peer Review File · Nature Communications]

DYRK1A signalling synchronizes the mitochondrial import pathways for metabolic rewiringREVIEWER COMMENTS

Reviewer #1 (Remarks to the Author):

Mitochondria have to import hundreds of different precursor proteins from the cytosol. The TOM complex of the outer membrane serves as universal entry gate for almost all precursor proteins. The mechanisms that regulate the targeting to and translocation through the TOM complex are only poorly understood. The Meisinger laboratory identified cytosolic kinases and phosphatases as key factors in this respect. By phosphorylation of cytosol-exposed domains of TOM receptors they adapt the import system to the prevailing metabolic conditions. DYRK-1-mediated phosphorylation of Tom70 represents a well-characterized example for such a modification.

A recent study claimed that Tom70 was phosphorylated by CK2 and that this modification would diminish the import of the IMS protein Mic19. In the present study, Meisinger and colleagues now present convincing evidence that both is presumably wrong: neither is the S94 site of Tom70 a target of CK2 (but rather of DYRK-1), nor does the IMS protein Mic19 use Tom70 as an import receptor.

The study by the Meisinger lab is of very high technical quality and all points raised are certainly valid. Since this is a topic of very high general relevance and importance for the field, a publication in Nature Communications is certainly justified, even if most of the data are negative and mainly intended to correct published data.

Nevertheless, the authors should tone down some of their statements about the Latorre-Muro et al. study and avoid the rebuttal character of the manuscript. In order to be published in Nature Communication, the study should contain scientific information about DYRK-1, TOM70 or MIC19 that is of relevance for a broad readership, independent of the Latorre-Muro et al. publication. Thus, if the authors can add more general information, this study is certainly suited for Nature Communications.

Specific points

1. The authors should tone down the statements about the poor quality and uncritical nature of the Latorre-Muro et al. publication. They still can make their point of doubt very clear.
2. The authors need to add some 'positive' information that is independent of the correction of a previous study. For example, they could use proteomics to directly compare the proteomes of phosphomimetic and phosphodeficient Tom70 and Tom20 mutants. Even if some of this information was published before, a direct comparison could certainly support the notion that MIA substrates such as MIC19 are not dependent on TOMM70 in vivo and, more important, provide a dataset of general relevance for a broad readership. This is particularly interesting because previous measurements were not made in the interesting brown adipose tissue model that was used here.

3. The authors use an in vitro system for their MIC19 import experiments. Since they make strong statements about the irrelevance of TOMM70 for the MIC19 import, it will be important to show in vivo data. In vivo pulse chase labeling experiments which follow MIA-dependent MIC19 oxidation would be one option here but other assays certainly might also be suited.

4. The authors claim that MIC19 is imported in an TOMM20-mediated manner. The main evidence comes from in vitro binding data to a purified TOMM20 domain. Previous studies by several laboratories came to the conclusions that MIA substrates do not bind to the cytosol-exposed TOM receptors of Tom20, Tom22 and Tom70 with any considerable affinity. The authors should therefore provide more compelling data about the role of TOMM20 for MIC19 import or be more cautious in the wording.

Reviewer #2 (Remarks to the Author):

Marada and colleagues provide a very interesting work which emerges as a response to discrepancies between some of their previous observations (that DYRKA1 phosphorylates TOM70 and facilitates this way the transport of TOM70-bound precursors to the import pore) and a parallel manuscript published in Cell Metabolism (Latorre-Muro et al.) suggesting that this very same phosphorylation acts oppositely.

By tackling this contradiction, the authors experimentally dismantle most of the main claims from the Cell Metabolism paper. The experiments are very clear and to the point, leaving little doubt that CK2 does not phosphorylate TOM70 and that TOM70 is not directly responsible for the mitochondrial import of MIC19. Reconciling some of the observations between the two papers, the authors propose that CX4945, an CK2 inhibitor used in the Latorre-Muro paper, also acts as a DYRKA1 inhibitor. While the pieces fit together, it is also true that the effect of CX4945 on DYRKA1 had been already reported.

Collectively, the manuscript is brilliant and helps clearing the field in a very conclusive way. In this sense, the authors need to be congratulated for their dedication and efforts. One always wishes more works like this were published when needed. If I had to be critical, my only point is that the authors could have spread a bit further than a direct rebuttal to the Cell Met paper in conceptual/novelty terms.

Some comments can be found below, which I hope the authors find useful:

1) In Figure 1, the treatment with NE lasts for 18 hours. I understand that the authors simply tried to reproduce the experiment from Latorre-Muro. Yet, as they might appreciate, this timing is just far too long to evaluate a direct signaling events derived from NE and probably catches indirect/secondary effects. In order to fully rule out an effect of NE signaling on TOM70 phosphorylation, could the authors try some shorter timings (minutes/hours)?

2/ The results also demonstrate that TOM70 is constitutively phosphorylated (at least in their working model). Nevertheless, there might be a very slight effect in Fig.1B (let panel). Is this something that could be quantified?

3/ In Figure 1C, could the authors use a positive control (this is, a protein whose abundance changes in response to NE at this timing? - Maybe UCP1?)

4/ In relation to Figure 4E, the authors claim that MIC19 import requires TOM20. While the result leave no doubt that TOM20 might enhance its import, I am not fully convinced that this is the only way MIC19 used to enter the mitochondria. Could the authors clarify this point?

5/ In Figure 6E, I honestly fail to see the changes claimed by the authors in TOM20 or TOM40 upon CX or INDY treatment. Could these changes be quantified to have a clearer idea of their magnitude?

6/ Also in Figure 6E, could the authors measure TOM70 and TOM20 phosphorylation?

7/ As a general note, I find it more practical when the experimental timings are explicitly mentioned in the figure legends and not just in the methods (at least when it comes to the exposure of cells to inhibitors or treatments).

REVIEWER COMMENTS

Reviewer #1 (Remarks to the Author):

Mitochondria have to import hundreds of different precursor proteins from the cytosol. The TOM complex of the outer membrane serves as universal entry gate for almost all precursor proteins. The mechanisms that regulate the targeting to and translocation through the TOM complex are only poorly understood. The Meisinger laboratory identified cytosolic kinases and phosphatases as key factors in this respect. By phosphorylation of cytosol-exposed domains of TOM receptors they adapt the import system to the prevailing metabolic conditions. DYRK-1-mediated phosphorylation of Tom70 represents a well-characterized example for such a modification.

A recent study claimed that Tom70 was phosphorylated by CK2 and that this modification would diminish the import of the IMS protein Mic19. In the present study, Meisinger and colleagues now present convincing evidence that both is presumably wrong: neither is the S94 site of Tom70 a target of CK2 (but rather of DYRK-1), nor does the IMS protein Mic19 use Tom70 as an import receptor.

The study by the Meisinger lab is of very high technical quality and all points raised are certainly valid. Since this is a topic of very high general relevance and importance for the field, a publication in Nature Communications is certainly justified, even if most of the data are negative and mainly intended to correct published data.

> We thank the reviewer for this very positive comment and appreciation of our clarification of this conflict in the field.

Nevertheless, the authors should tone down some of their statements about the Latorre-Muro et al. study and avoid the rebuttal character of the manuscript. In order to be published in Nature Communication, the study should contain scientific information about DYRK-1, TOM70 or MIC19 that is of relevance for a broad readership, independent of the Latorre-Muro et al. publication. Thus, if the authors can add more general information, this study is certainly suited for Nature Communications.

> We have revised and changed the main focus of our manuscript so that the different TOM70 phosphorylation models now provide the starting point for an in-depth molecular analysis of the regulation of the TOM complex by cytosolic kinases. For this, we have combined the figures related to findings from Latorre-Muro et al and moved several experiments to the supplement. Next, we added several new experiments so that the focus of the manuscript shifted towards novel scientific findings: We find that not only MIC19 import into the intermembrane space but also import of a classical presequence precursor protein into the matrix is stimulated upon chemical inhibition of DYRK1A (see **new Figure 6a, b** with the model matrix precursor HSPD1). Thus, remodelling of the TOM complex upon DYRK1A inhibition (via subsequent transcriptional response) does not only re-balance the metabolite carrier import pathway (to compensate loss of TOM70 phosphorylation), but also stimulates the presequence import pathway via upregulation of the presequence import receptors. Thus, DYRK1A emerges as central regulatory platform for maintaining and synchronizing the mitochondrial import pathways (**new Figure 6c**). This is an exciting novel concept in the field of mitochondrial biogenesis and of relevance for a broad readership.

Moreover, we show now that CHCHD6/MIC25, a further IMS protein and MIA40 substrate, requires TOM20 (and not TOM70) as import receptor. Therefore, our findings point to a more general role of TOM20 as an import receptor for human intermembrane space proteins.

The new concept, that DYRK1A serves as regulatory platform synchronizing the different mitochondrial import routes, may have important implications in the understanding of pathomechanisms in which both, loss of DYRK1A function (causing e.g. microcephaly, autism spectrum disorder, or developmental delay) as well as DYRK1A gain of function (e.g. Down syndrome as DYRK1A is encoded on chromosome 21) can impact on mitochondrial metabolism. We included this aspect in the discussion part of the revised manuscript.

New Figures 6c: Schematic showing the role of DYRK1A as central regulatory hub to link the main mitochondrial import routes. While DYRK1A is essential to activate the carrier import pathway via TOM70^{pSer91} phosphorylation (left panel), its impairment leads to a transcriptional activation of all three TOM import receptors followed by remodelling of the TOM complex that allows an activation of the MIA- and the presequence import pathways (right panel).

Specific points

1. The authors should tone down the statements about the poor quality and uncritical nature of the Latorre-Muro et al. publication. They still can make their point of doubt very clear.

> We have rephrased several of our statements to tone down the rebuttal character of our manuscript as requested by the reviewer. Moreover, we have combined Figures 1 and 2 which were directly related to the Latorre-Muro publication to a single Figure and transferred further experiments to the new supplementary Figure 1. This allowed us to focus now in the main figures on several new results including the analysis of the import route for a second MIA substrate (CHCHD6) and its dependency on the receptor TOM20 (see **new Figures 3g and 4d,g**) and, even more exciting, the impact of DYRK1A inhibition on the main presequence import pathway (see **new Figure 6**).

2. The authors need to add some 'positive' information that is independent of the correction of a previous study. For example, they could use proteomics to directly compare the proteomes of phosphomimetic and phosphodeficient Tom70 and Tom20

mutants. Even if some of this information was published before, a direct comparison could certainly support the notion that MIA substrates such as MIC19 are not dependent on TOMM70 *in vivo* and, more important, provide a dataset of general relevance for a broad readership. This is particularly interesting because previous measurements were not made in the interesting brown adipose tissue model that was used here.

> As described above, we now show that inhibition of DYRK1A stimulates the major and central presequence import pathway into mitochondria (**new Figure 6**; shown for the matrix-targeted precursor protein HSPD1). DYRK1A therefore emerges as central regulator for synchronizing the main import pathways into mitochondria. We believe that this is a new concept in regulation of mitochondrial protein biogenesis that might have important implications e.g. for the understanding of DYRK1A-related pathologies as both, loss of functions (e.g. DYRK1A syndrome or autism) but also gain of function of DYRK1A (e.g. Down syndrome) signalling could affect the mitochondria-DYRK1A axis.

In addition and confirming our MIC19 data, we now show, that import of a further IMS protein, CHCHD6, is independent on TOM70 but requires TOM20 (import experiments into isolated mitochondria from TOM70 and TOM20 KD cells (see **new Figure 3g**) and CHCHD6 precursor binding to the purified TOM20 and TOM70 receptor domains (**new Figure 4d,g**)).

New Figures 6a and 6b: Import of matrix targeted HSPD1 precursor into isolated mitochondria from HEK293T cells incubated in the presence or absence of INDY or CX4945. Both inhibitors lead to activation of HSPD1 import.

New Figure 3g: Import of radiolabelled MIA substrate CHCHD6 into isolated mitochondria from TOM70, TOM20 or control (Ctrl.) knockdown cells.

New Figure R1: Western blot analysis of mitochondria isolated from TOM20, TOM70 and control (Ctrl.) knockdown cells. siRNAs treatment for 48h. VDAC3 serves as loading control.

3. The authors use an in vitro system for their MIC19 import experiments. Since they make strong statements about the irrelevance of TOMM70 for the MIC19 import, it will be important to show in vivo data. In vivo pulse chase labeling experiments which follow MIA-dependent MIC19 oxidation would be one option here but other assays certainly might also be suited.

> Our findings that MIC19 import does depend on the import receptor TOM20 but not TOM70 is based in part on in vitro binding of the precursor to purified receptor domains (Figure 3) as stated by the reviewer. In addition, we find that MIC19 import is severely reduced into isolated intact mitochondria from TOM20 knockdown cells, while TOM70 knockdown has no effect (Figure 2). Moreover, we show that TOM20 antibodies but not TOM70 antibodies can compete with the import of MIC19 in organello (Supplementary Figure 2). Thus, our findings are based on three independent experimental approaches. With CHCHD6 we now show for a further IMS precursor protein with the characteristic twin CX₉C motif of MIA substrates a dependency on TOM20 but not TOM70 (see response point 2 above). This result strongly supports our findings that MIC19 import is independent on TOM70 but needs TOM20. Moreover, this might also indicate a more common import mechanism for IMS proteins in human mitochondria.

In addition, we followed the suggesting of the reviewer to perform an in vivo pulse chase labelling experiment followed by assessment of MIA dependent oxidation. While we successfully could set up such a system and could monitor the oxidation of MIC19 in vivo and gain an in vivo import kinetics (see Fig. R2) the application of this assay to TOM20 and TOM70 KD cells was not possible as the labelling efficiency with [³⁵S] methionine/cysteine was strongly reduced in the receptor KD cells compared to control cells (see Fig. R3).

New Figure R2: Pulse chase in vivo import experiment in the presence of [³⁵S]methionine/cysteine. Pulse was for 5 min. followed by chase for up to 8 min. **Top panel:** pellet fractions containing mitochondria were isolated and imported MIC19 protein immunoprecipitated and detected via autoradiography. **Bottom panel:** Same reaction but samples were treated with AMS (allyl methyl sulfide) as alkylation reagent to monitor formation of disulfide bonds (oxid.) upon import into the IMS.

New Figure R3: In vivo labelling of cells after knockdown using TOM20 or TOM70 or control siRNA. Cells were incubated in the presence of [³⁵S]methionine/cysteine and analyzed via SDS-PAGE and autoradiography.

4. The authors claim that MIC19 is imported in an TOMM20-mediated manner. The main evidence comes from in vitro binding data to a purified TOMM20 domain. Previous studies by several laboratories came to the conclusions that MIA substrates do not bind to the cytosol-exposed TOM receptors of Tom20, Tom22 and Tom70 with any considerable affinity. The authors should therefore provide more compelling data about the role of TOMM20 for MIC19 import or be more cautious in the wording.

> We agree with the reviewer that the previous view in the field was rather that MIA substrates are imported independent from the import receptors. Notably, most of these studies were performed in the yeast model, while only little has been investigated here in human mitochondria (at least to our knowledge).

As pointed out already above, our conclusion that MIC19 requires TOM20 is not only substantiated by in vitro binding data but also by in organello import into TOM20 and TOM70 knockdown mitochondria as well as by competition assays using TOM20 and TOM70 antibodies in wildtype mitochondria. The requirement of import receptors at all is also reflected by the inhibition of MIC19 import via prior removal of surface receptors with trypsin (Figure 2a, b and Supplementary Figure 2a). In addition, our findings are now further substantiated by testing of an additional MIA substrate, CHCHD6, which harbors the canonical twin CX₉C motif like MIC19. CHCHD6 import is, like MIC19, delayed in TOM20 KD mitochondria but not in TOM70 KD (see **new Figure 3g**). In vitro binding assays further demonstrates binding of CHCHD6 precursors to the TOM20 but not TOM70 receptor domain (**new Fig 4d,g**).

Thus, our new data confirm this import pathway also for a second MIA substrate. We speculate now that this might represent a common pathway in human mitochondria.

New Figure 4d: Analysis of elution fraction from binding assay for radiolabelled CHCHD6 precursor protein to immobilized cytosolic receptor domains of human TOM70 and TOM20. CHCHD6 precursor binds more to TOM20_{cd} than TOM70_{cd} similar like MIC19 and OTC while the carrier substrate TIM23 preferentially binds TOM70. See Fig. 4g for quantification.

New Figure 4g: Quantification of CHCHD6 receptor binding in comparison to MIC19, OTC and TIM23.

Reviewer #2 (Remarks to the Author):

Marada and colleagues provide a very interesting work which emerges as a response to discrepancies between some of their previous observations (that DYRKA1 phosphorylates TOM70 and facilitates this way the transport of TOM70-bound precursors to the import pore) and a parallel manuscript published in *Cell Metabolism* (Latorre-Muro et al.) suggesting that this very same phosphorylation acts oppositely.

By tackling this contradiction, the authors experimentally dismantle most of the main claims from the *Cell Metabolism* paper. The experiments are very clear and to the point, leaving little doubt that CK2 does not phosphorylate TOM70 and that TOM70 is not directly responsible for the mitochondrial import of MIC19. Reconciling some of the observations between the two papers, the authors propose that CX4945, an CK2 inhibitor used in the Latorre-Muro paper, also acts as a DYRKA1 inhibitor. While the pieces fit together, it is also true that the effect of CX4945 on DYRKA1 had been already reported.

Collectively, the manuscript is brilliant and helps clearing the field in a very conclusive way. In this sense, the authors need to be congratulated for their dedication and efforts. One always wishes more works like this were published when needed. If I had to be critical, my only point is that the authors could have spread a bit further than a direct rebuttal to the *Cell Met* paper in conceptual/novelty terms.

> We thank the reviewer for this very positive comment and appraisal of our clarification of this contradictory findings.

Some comments can be found below, which I hope the authors find useful:

1) In Figure 1, the treatment with NE lasts for 18 hours. I understand that the authors simply tried to reproduce the experiment from Latorre-Muro. Yet, as they might appreciate, this timing is just far too long to evaluate a direct signaling events derived from NE and probably catches indirect/secondary effects. In order to fully rule out an effect of NE signaling on TOM70 phosphorylation, could the authors try some shorter timings (minutes/hours)?

> We thank the reviewer for pointing this out. We have incubated the PIBA cells now for shorter times (30 and 120 min) with NE and did not detect any changes in the level of TOM70 phosphorylation (see new Figure R2).

New Figure R2: Analysis of TOM70^{Ser91} phosphorylation (using phospho specific antibody) upon short time incubation of differentiated PIBA cells with NE for 30 and 120 min. VDAC3 serves as loading control.

2/ The results also demonstrate that TOM70 is constitutively phosphorylated (at least in their working model). Nevertheless, there might be a very slight effect in Fig.1B (let panel). Is this something that could be quantified?

> We have quantified the ratio of phosphorylated vs non-phosphorylated TOM70 upon NE stimulation from PhosTag gels of three independent experiments (see Figure 1B as representative experiment). The ratio of TOM70 Ser91 phosphorylation and non-phosphorylated TOM70 does not change significantly upon NE treatment (see new Fig. 1c).

New Figure 1c: Quantification of PhosTag gels from Figure 1 B reveal no significant changes upon NE treatment.

3/ In Figure 1C, could the authors use a positive control (this is, a protein whose abundance changes in response to NE at this timing? - Maybe UCP1?)

> We have tested UCP1 as well as eIF2 α ^{PS51} protein levels in cells that were used for the analysis in Figure 1C and find an upregulation of both markers for NE treatment as expected.

New Figure R3: Cell extracts from differentiated PIBA cells were incubated for 18h in the presence or absence of NE (experimental design as in Figure 1 and Latorre-Muro et al., 2021). Samples were then analyzed with UCP1 and eIF2 α ^{PS51} antibodies; VDAC3 serves as loading control.

4/ In relation to Figure 4E, the authors claim that MIC19 import requires TOM20. While the result leave no doubt that TOM20 might enhance its import, I am not fully convinced that this is the only way MIC19 used to enter the mitochondria. Could the authors clarify this point?

> As pointed out above, our finding that MIC19 import requires TOM20 is underpinned by three independent experimental approaches (in vitro precursor binding to TOM20 receptor domain, in organello import into TOM20 knockdown mitochondria and competition of its import with TOM20 antibodies in wildtype mitochondria (Figures 2, 3 and Supplementary Figure 2). Together with the new experiments showing very similar dependency for a further MIA substrate (CHCHD6), we are confident that human MIA substrates interact and require the TOM20 receptor for mitochondrial import. However, and this is a valid point of the reviewer, TOM20 cannot mediate the entire import pathway for MIA substrates alone. We expect that only together with the central import receptor TOM22 which is essential and involved in basically every import route into mitochondria, TOM20 bound precursor can be transferred to the import translocon. As this is also the case for the TOM70 dependent precursors, we performed our assays under conditions in which TOM22 protein levels were actually not affected. This allows us to draw specific and differentiated conclusions on the individual functions of TOM20 and TOM70 for the import of MIA substrates as we do not risk secondary effects due to loss of TOM22 (see e.g. Figures 3a and 3c in which TOM22 remains unaffected). In addition, we have now quantified the knockdown efficiencies for TOM20 and TOM70 and find that TOM70 which we propose to be not an import receptor for MIC19 and CHCHD6 is even more strongly reduced than TOM20, which we find as the actual MIC19 and CHCHD6 receptor (see **new Figure 3d**). Furthermore, the residual TOM20 left in TOM20 KD mitochondria might still be sufficient to allow import in principle but slows down the efficiency due to more limited access of precursor to the import receptor.

New Figure 3d: Quantification of knockdown efficiency for TOM20 and TOM70 via siRNA treatment for 48h.

5/ In Figure 6E, I honestly fail to see the changes claimed by the authors in TOM20 or TOM40 upon CX or INDY treatment. Could these changes be quantified to have a clearer idea of their magnitude?

> We have quantified the western signal from Figure 6E (now Figure 5e) and can confirm the significant upregulation of the TOM import receptors upon INDY and CX4945 treatment while the central import pore TOM40 and the loading control GRP75 remain unaffected, fully supporting our findings.

New Figure 5f: Quantification of western blots from Figure 6E confirms significant increase of all three import receptors while the level of the import pore TOM40 remains largely unaffected. GRP75, control protein.

6/ Also in Figure 6E, could the authors measure TOM70 and TOM20 phosphorylation?

> We have analyzed TOM70 and TOM20 phosphorylation of samples from Figure 6E via PhosTag gels (**new Figure R4**). For both TOM receptors we see the expected increase of protein levels upon CX4945 and INDY treatment. For TOM20 we don't see additional bands that would indicate phosphorylation. For TOM70 we observe the expected change in the ratio of phosphorylated vs. non-phosphorylated TOM70 that allows to maintain the carrier import pathway in agreement with our previous results (Walter et al., 2021).

New Figure R4: PhosTag analysis of samples from CX4945 and INDY treated HEK293T cells analyzed with TOM20 and TOM70 antibodies.

7/ As a general note, I find it more practical when the experimental timings are explicitly mentioned in the figure legends and not just in the methods (at least when it comes to the exposure of cells to inhibitors or treatments).

> We thank the reviewer for this very important point. We have now added the experimental timings for all incubations/treatments of cells in the respective figure legends.

REVIEWERS' COMMENTS

Reviewer #1 (Remarks to the Author):

The authors satisfactorily addressed all points raised on the previous version. I fully support publication of this exciting study in its present form. It convincingly describes an interesting mechanism by which cells direct precursor proteins onto different mitochondrial import pathways.

Reviewer #2 (Remarks to the Author):

The authors adequately addressed all my comments. I can only commend them for a very elegant and thorough work.